# TRA: Better Length Generalisation with Threshold Relative Attention

**Mattia Opper**                                                          *m.opper@ed.ac.uk*
*University of Edinburgh**

**Roland Fernandez**                                                *rfernand@microsoft.com*
*Microsoft Research*

**Paul Smolensky**                                                      *psmo@microsoft.com*
*Microsoft Research*
*Johns Hopkins University*

**Jianfeng Gao**                                                        *jfgao@microsoft.com*
*Microsoft Research*

**Reviewed on OpenReview:** *https://openreview.net/forum?id=yNiBUc2hMW*

## Abstract

Transformers struggle with length generalisation, displaying poor performance even on basic tasks. We test whether these limitations can be explained through two key failures of the self-attention mechanism. The first is the inability to fully remove irrelevant information. The second is tied to position, even if the dot product between a key and query is highly negative (i.e. an irrelevant key) learned positional biases may unintentionally up-weight such information - dangerous when distances become out of distribution. Put together, these two failure cases lead to compounding generalisation difficulties. We test whether they can be mitigated through the combination of a) **selective sparsity** - completely removing irrelevant keys from the attention softmax and b) **contextualised relative distance** - distance is only considered as between the query and *the keys that matter*. We show how refactoring the attention mechanism with these two mitigations in place can substantially improve the generalisation capabilities of decoder only transformers.[1]

## 1 Introduction

Transformers (Vaswani et al., 2017) have sparked a revolution in artificial intelligence. When pre-trained at scale, these models display impressive broad-spectrum capabilities, particularly in the domains of natural language and code (Bubeck et al., 2023; Dubey et al., 2024). This includes the ability to perform complex abstract symbolic processing through in-context learning (ICL) (Brown et al., 2020; Sinha et al., 2024; Smolensky et al., 2024). On the other hand, these abilities are far from robust (Dziri et al., 2023; Mirzadeh et al., 2024) and display dramatic failure modes such as self-contradiction and hallucination (Liu et al., 2023a; Ji et al., 2023; Guerreiro et al., 2023). Similar failure modes are also observed in controlled synthetic settings where the correct solution *should be trivial*. Decoder-only transformers consistently display poor generalisation to unseen lengths and dependency distances (Zhou et al., 2023; Liu et al., 2023a).

Addressing these issues can provide crucial benefits regarding both long context utilisation and length generalisation more broadly. Firstly, current approaches generally involve training at a certain sequence length and then adapting the positional encoding to longer context. Most commonly this entails additional fine-tuning for best effect, thereby incurring a further compute cost penalty. Even then current SoTA foundation

---

*Work completed while at Microsoft Research
[1] Pytorch code for core mechanism in Appendix A.

models still struggle with long context (Hsieh et al., 2024). Ideally, we would want a solution that simply works out the box. Put another way, if the transformer is in fact learning circuits (Merrill et al., 2022; Olsson et al., 2022; Liu et al., 2023b), then their correct execution should be general purpose and robust, rather than contingent on specific quirks of the training data.

To understand how to get there, we should first look at the core capability that the attention mechanism was supposed to enable in the first place, namely the ability to selectively retrieve and propagate information (Bahdanau et al., 2014; Luong et al., 2015; Vaswani et al., 2017). We can break this capability down into three broad categories. (1) Purely positional: attending to ones local neighbourhood (i.e. contextualisation within a sentence or phrase). (2) Purely based on content: attending to the most relevant item irrespective of position. (3) Content followed by position: of the relevant items apply a positional bias (e.g. select the most recent). (3) is a crucial capability, it enables tracking the state of an entity over time, and is a pre-requisite for robust reasoning. Our contention is that standard attention mechanisms can perform (1) and (2), but struggle with (3). This due to the way that semantic content and positional information combine. Position is generally treated as independent of content. Which means that favouring one must come at the expense of the other. This leaves the two types of information in conflict, fundamentally limiting self-attention.

Our solution involves a two step process. We first mask out irrelevant keys based on threshold applied to the raw attention weights, and then compute the relative distance but only as between the query and the keys that remain. We call this mechanism **T**hreshold **R**elative **A**ttention (**TRA**). TRA is a simple, modular addition that allows self-attention to learn general solutions to all three behaviours outlined above. We find that TRA's introduction leads to considerably improved length generalisation on synthetic benchmarks. As well as robust, and even improved, language modeling perplexity for out-of-distribution sequence lengths, even up to an increase of 32x. Our findings contribute to the development of more robust attention mechanisms.

## 2 Model

### 2.1 Background: Relative Positional Encodings

Relative positional encodings (Dai et al., 2019; Raffel et al., 2019) decompose the attention operation into two steps. First the standard $QK^T$ dot product is calculated, but this purely based on semantic relevance. Let's call that matrix $S$ (semantic). This is then added with a learned positional bias matrix $D$ (distance) to produce the final attention weight.

$$S = \frac{QK^T}{\sqrt{d_k}} \tag{1}$$

$$A = \text{softmax}\left(S + D\right) \tag{2}$$

Our contention is that a central cause of length generalisation issues stems from $S$ and $D$ being treated as independent of each other. If $D$ is fit to the training distribution, then distribution shifts will necessarily begin to introduce errors. $S$ is more robust in this regard, because relevance isn't impacted by changing lengths. We want to make $D$ more robust by making it contingent on $S$.

### 2.2 Contextual Relative Distance

We want to condition $D$ on $S$. In other words we only want to consider position over keys that matter. To achieve this we first introduce **selective sparsity**. We will allow certain keys to be fully removed from $S$, but do not force it to be one hot. Instead we introduce a **threshold** and eliminate all keys which fall below it. Lets call that matrix $S'$. The simplest way to get $S'$ is by applying a ReLU.

$$S' = \text{ReLU}\left(\frac{QK^T}{\sqrt{d_k}}\right) \tag{3}$$

Next calculate a boolean mask $\boldsymbol{M}$ that is true only for relevant positions:

$$M_{i,j} = \begin{cases} 1 & \text{if } S'_{i,j} > 0 \\ 0 & \text{otherwise} \end{cases} \tag{4}$$

We are now going to compute our contextualised distance matrix, but only consider the relative distance to keys which make it through the threshold, this give us the **contextualised relative distance**. To do this we consider the following example $\boldsymbol{M}$, where again only keys which meet the threshold are assigned 1:

$$\boldsymbol{M} = \begin{bmatrix} 1 & 0 & 0 & 0 \\ 1 & 0 & 0 & 0 \\ 0 & 1 & 1 & 0 \\ 1 & 0 & 1 & 0 \end{bmatrix} \tag{5}$$

Our threshold mask $\boldsymbol{M}$ is a $m \times n$ matrix, which we are going to use to get our contextualised relative distance matrix $\bar{\boldsymbol{D}}$, by applying the cumsum operation in the right to left direction (i.e. producing relative positions given causal masking). More formally we can say that for row index i and column index j:

$$\bar{D}_{ij} = \sum_{k=j}^{n} M_{ik} \tag{6}$$

For our example $\boldsymbol{M}$ from (5) this would yield the following output, noting that we omit the keys which do not meet the threshold in this example for clarity:

$$\bar{\boldsymbol{D}} = \begin{bmatrix} 1 & 0 & 0 & 0 \\ 1 & 0 & 0 & 0 \\ 0 & 2 & 1 & 0 \\ 2 & 0 & 1 & 0 \end{bmatrix} \tag{7}$$

$\bar{\boldsymbol{D}}$ now contains distances which are informed by context. For example, for the query in position two it views the key in position one as the closest possible item rather than its own key, which has been deemed irrelevant through the threshold operation. Similarly, for the query at position 4 it will view the key at position three as the closest possible item and the key at position 1 as the next nearest, skipping over the irrelevant intervening keys. This enables operation (3) from the introduction i.e. considering ordering but only over the relevant items. A capability which more traditional PE methods such as RPE (Dai et al., 2019), absolute (Vaswani et al., 2017) and RoPE Su et al. (2021) are unable to accomodate because they treat position as a *fixed property* rather than as adaptive to the context dependent needs of the query.

## 2.3 Parameterising $\bar{D}$

Now we have arrived at our contextualised relative distances, but we still need some way of turning them into an actual weight. In order to do so we utilise a parameterised forget gate. Lets denote the residual stream of size $E$ for position $i$ as $\boldsymbol{x}_i$, with $\boldsymbol{W}_f \in \mathbb{R}^{E \times 1}$ as the forget projection, $b$ as a scalar bias, and $\sigma$ as the sigmoid non-linearity. Then the weight for each position is given as:

$$\delta_i = \sigma\left(W_f x_i + b\right) \tag{8}$$

$$D'_{ij} = \delta_i^{\bar{D}_{ij}} \tag{9}$$

As each $\delta_i$ is a sigmoided scalar value, the more times it is multiplied with itself the smaller it gets. The closer $\delta_i$ is to 1 the longer the decay takes, the closer to zero the shorter. This provides the model a temporal

memory. Enabling it to forget irrelevant past information if necessary, or to treat all timesteps uniformly if temporal ordering is not a concern. We opt for a recency bias for several reasons. Firstly, it has been generally shown to be useful when modeling natural language (Shen et al., 2018; Zhu et al., 2021; Clark et al., 2025). In particular ALiBi (Press et al., 2021) shows that a non-parametric recency bias significantly improves length generalisation in transformers. However, though their non-parametric version makes this constraint rigid. By choosing to parameterise it instead we allow the model to completely disregard position if needed, the benefits of which are demonstrated by Lin et al. (2025). Secondly, it is unclear whether a token occupying the ith position provides useful information beyond causal ordering. The first occurrence can be identified by proximity to the start of sequence token. While middle positions are better differentiated via their unique semantic context, a capability which has been identified and carefully studied in transformers by Ebrahimi et al. (2024). Finally, a temporally decaying memory is a powerful mechanism. Merrill et al. (2024) show that is better able to approximate sophisticated $NC^1$ circuits given finite depth, compared to regular transformers. Moreover, it is the mechanism which has empowered the resurgence in state space models (Peng et al., 2023; Orvieto et al., 2023; De et al., 2024), recent successful recursive networks (Opper & Siddharth, 2024a), and has shown considerable promise with transformers (Csordás et al., 2021; Lin et al., 2025).

### 2.4 The Final Attention Weights

We compute the final attention weights $\boldsymbol{A'}$:

$$\hat{\boldsymbol{A}} = \boldsymbol{S'} + \log(\boldsymbol{D'}) \tag{10}$$

$$\bar{A}_{ij} = \begin{cases} -1e11 & \text{if } M_{ij} = 0 \\ \hat{A}_{ij} & \text{otherwise} \end{cases} \tag{11}$$

$$\boldsymbol{A'} = \text{softmax}\left(\bar{\boldsymbol{A}}\right) \tag{12}$$

This gives us the full TRA mechanism. We use $\log(\boldsymbol{D'})$ for improved numerical stability and weight scale following Lin et al. (2025). Note keys which do not meet the threshold are completely removed from the final softmax by setting them to a large negative value (-1e11), as well as not counting towards distance as dicussed previously. This means that TRA exhibits both selective sparsity and allows semantic content to determine, and consequently synergise with, positional bias.

## 3 Experiments

### 3.1 Tasks

Our core hypothesis is that length generalisation failures occur in transformers because the standard attention mechanism puts semantic and positional information in conflict. TRA is designed to mitigate this. To test our hypothesis, we first turn to controlled synthetic tasks, followed by language modeling.

**Flip-Flop Language Modelling:** Introduced by Liu et al. (2023a) Flip-Flop language modelling is a algorithmic reasoning benchmark designed to test transformers ability to glitchlessly handle sequential dependencies. Sequences consist of symbol alphabet of three instructions: write (w), ignore (i) and read (r) each of which is followed by a bit. To solve the task the model has to recall the bit that follows the nearest write instruction while ignoring all irrelevant information (e.g. in: w1i0i1i1i0i1i0i0i1i0r out: 1). Memorising sequences will lead to failure. The model has to learn a program to succeed. The ability to model a flip-flop is the foundation for all syntactic parsing and algorithmic reasoning capabilities (it necessitates the propagation of information according to abstract rules), and is a pre-requisite to being able to maintain a state hierarchically (Merrill et al., 2022; Liu et al., 2023b). The task consists of three test sets: IID, and two out of distribution sets that vary the number of intervening ignore instructions. OOD sparse increases the number of ignore instructions and requires the ability to handle increased dependency distance. OOD dense decreases the number of ignore instructions and consequently probes whether the model retain focus in the presence of an increased number of attractors (i.e. write instructions). Input examples consist of strings of length 512 with the probability of an ignore instruction being generated set to 0.98 in the sparse set and 0.1

in the dense set. The criterion for success is perfection (i.e. 100% accuracy across all test sets). The authors find that transformers exhibit reasoning errors across a wide range of architectural variants and that the issue is *scale-invariant*, persisting even up to GPT-4.

**Induction Heads:** This is a test of associative memory. Given a string of unique symbols the model must learn the following reasoning pattern: a → b ... a → ? out: b, otherwise known as an induction head. This ability has been shown to emerge for specific heads and for specific input-output pairs after exposure to roughly two billion tokens of pre-training data (Olsson et al., 2022), and is hypothesised to be a crucial mechanism for in-context learning. However, when tested in controlled synthetic settings, transformers' ability to learn the generalised pattern has been limited. Zhou et al. (2023) show that models are only able to weakly generalise (from length ≈40 to 50) despite the fact that they *should* in theory easily be able to learn the circuit. Beyond basic positional reasoning the induction heads task is also a test of the attention mechanisms capacity for specificity. As length increases the number of irrelevant features grows and the model must be able to sharply discriminate so as to select the correct successor.

**Copy:** A more difficult version of the induction heads task is copying non-unique strings. Here we evaluate using a vocabulary of 10 symbols. The model must learn to absorb the history for each token in order to predict the correct successor. In out of distribution length settings difficulty is increased because the level of history required to distinguish between potential successors increases. It is also a task where our current models of transformer programs, which treat symbol manipulation as fully discrete (i.e. one-hot selection), cannot find a solution (Weiss et al., 2021; Zhou et al., 2023), and therefore predict generalisation failure. Copying is a test of the models capacity to maintain a memory. A given tokens capacity to memorise its past determines successful disambiguation of its successor.

For all tasks we use full sequence accuracy as the criterion (exact match). For the copy and induct tasks we train on sequences with input length 50 and evaluate on buckets of increased OOD lengths up to 300.

## 3.2 Baselines and Experimental Setup

For both TRA and the baselines we use the same core transformer backbone based on the Llama 3 variant of the architecture (Dubey et al., 2024); consisting on SwiGLU layers Shazeer (2020) coupled with RMSNorm (Zhang & Sennrich, 2019). We compare TRA with the following baselines:

**No positional encoding (NoPE):** Kazemnejad et al. (2023) claim that decoder only transformers are implicitly able to represent positional information and that explicitly encoding it is unnecessary. Our first baseline is therefore simply standard causal attention with no positional embedding.

**Absolute positional encoding (APE):** Positional information for each index is represented by a learned embedding which is added to the residual stream at layer zero. This was the standard in the original variant (Vaswani et al., 2017) as well as GPT-2 (Radford et al., 2019). Lengths not encountered during training will not have the corresponding embeddings trained.

**Relative Positional Encodings (REL):** Introduced by Transformer-XL Dai et al. (2019) and also adopted by T5 (Raffel et al., 2019). Relative Positional Encodings model position via a learnable additive bias term. Position is considered as to the relative distance of key to the query. OOD distances are all assigned the same learned value for maximum length.

**Rotational Positional Encoding (RoPE):** Introduced by Su et al. (2021). This approach models positional information by applying a rotation to the keys proportional to the distance from the query. Recent work by Barbero et al. (2025) demonstrates that rather than encoding a gradual distance based decay RoPE actually learns either fully positional attention (attend to predecessor or attend to first token) or full semantic attention. One coming at the expense of the other.

**Label Encoding (Label):** Introduced by Li & McClelland (2023) these seek to mitigate the OOD issues faced by APE by randomly selecting and then sorting indices from a range greater than sequence length. The authors demonstrate that this aides length generalisation under a limited OOD setting ($L25 \rightarrow 50$), with the contention that this is due to the encodings learning to respect relative ordering rather than absolute value. The same mechanism is also proposed in concurrent work by Ruoss et al. (2023).

**Contextual Positional Encoding (CoPE):** Perhaps conceptually the most related to TRA, CoPE (Golovneva et al., 2024) utilises spikes in the $S$ matrix in order to calculate positions. Tokens are assigned fractional positions by interpolating betweeen a fixed set of absolute embeddings. Unlike TRA it does not completely remove irrelevant items from the softmax and uses an absolute rather than a relative scheme. This incurs further computational cost because *each query in each head* needs to be matrix multiplied with a layer specific absolute positional embedding matrix. The authors attempt to mitigate this issue by limiting the number of absolute positions to a small number. TRA allows for similar behaviour but in a more computationally efficient manner, and allows removal of noise through its thresholding mechanism.

**Forgetting Transformer (FoT):** The recent Forgetting Transformer (FoT) (Lin et al., 2025) utilises a forget gate as its positional encoding. The crucial difference between it and TRA is that the forgetting transformer uses standard relative distance compared with TRA's contextualised alternative. This means that it can either enforce a recency bias or switch off any positional information completely. Crucially, what it cannot do is synergise the two (capability (3) from the introduction, which is precisely what TRA's contextualised relative distance enables. FoT therefore serves as the truest test as to whether the issues we hypothesise regarding position are valid.

**Differential Transformer (Diff):** Introduced by Ye et al. (2024) the differential transformer utilises a twin heads approach where an auxiliary attention head looks to perform noise reduction by down-weighting attention to irrelevant keys. It tests to what extent simply the ability to remove noise from the attention dot product is beneficial - without any further considerations regarding position.

**Hyper-parameters:** For all models we use the mini configuration from Turc et al. (2019): four heads, four layers and a 256 dimensional embedding size. The MLP is set 2x embedding size for both the linear and gate hidden units. Our focus is on small models following prior work (Zhou et al., 2023). Furthermore, attention glitches have been shown to persist at scale (Liu et al., 2023a), and the true solution for these tasks should not require additional layers. We use a linear warm-up for the first 5% of steps coupled with cosine decay. Other task and model specific hyper-parameters can be found in Appendix B.

### 3.3 Results

Table 1: Results represent the average across four random initialisations. Metric is full sequence accuracy (exact match). The numbers next to Copy and Induct represent input length ranges.

| Model | NoPE | APE | REL | RoPE | Label | FoT | Diff | CoPE | TRA |
|---|---|---|---|---|---|---|---|---|---|
| Flip Flop IID | $100 \pm 0.0$ | $100 \pm 0.0$ | $100 \pm 0.0$ | $100 \pm 0.0$ | $100 \pm 0.0$ | $100 \pm 0.0$ | $100 \pm 0.0$ | $100 \pm 0.0$ | $100 \pm 0.0$ |
| Flip Flop OOD (Dense) | $0.15 \pm 0.0$ | $27.38 \pm 13.1$ | $41.2 \pm 47.29$ | $100 \pm 0.0$ | $12.58 \pm 15.87$ | $100 \pm 0.0$ | $100 \pm 0.0$ | $100 \pm 0.0$ | $100 \pm 0.0$ |
| Flip Flop OOD (Sparse) | $99.97 \pm 0.1$ | $90.61 \pm 5.64$ | $70.48 \pm 10.92$ | $72.82 \pm 1.27$ | $86.93 \pm 9.93$ | $93.4 \pm 4.21$ | $77.8 \pm 6.73$ | $95.1 \pm 4.4$ | $100 \pm 0.0$ |
| Induct IID (0-50) | $100 \pm 0.0$ | $100 \pm 0.0$ | $100 \pm 0.0$ | $100 \pm 0.0$ | $99.96 \pm 0.04$ | $100 \pm 0.0$ | $100 \pm 0.0$ | $100 \pm 0.0$ | $100 \pm 0.0$ |
| Induct OOD (50-100) | $13.42 \pm 1.96$ | $0.0 \pm 0.0$ | $0.0 \pm 0.0$ | $7.85 \pm 11.87$ | $45.36 \pm 43.99$ | $15.00 \pm 2.83$ | $4.43 \pm 0.0$ | $20.42 \pm 2.74$ | $100 \pm 0.0$ |
| Induct OOD (100-200) | $0.0 \pm 0.0$ | $0.0 \pm 0.0$ | $0.0 \pm 0.0$ | $0.0 \pm 0.0$ | $0.0 \pm 0.0$ | $0.0 \pm 0.0$ | $0.0 \pm 0.0$ | $0.0 \pm 0.0$ | $99.90 \pm 0.0$ |
| Induct OOD (200-300) | $0.0 \pm 0.0$ | $0.0 \pm 0.0$ | $0.0 \pm 0.0$ | $0.0 \pm 0.0$ | $0.0 \pm 0.0$ | $0.0 \pm 0.0$ | $0.0 \pm 0.0$ | $0.0 \pm 0.0$ | $99.33 \pm 0.0$ |
| Copy IID (0-50) | $100 \pm 0.0$ | $100 \pm 0.0$ | $100 \pm 0.0$ | $100 \pm 0.0$ | $100 \pm 0.0$ | $100 \pm 0.0$ | $100 \pm 0.0$ | $100 \pm 0.0$ | $100 \pm 0.0$ |
| Copy OOD (50-100) | $12.43 \pm 1.66$ | $0.0 \pm 0.0$ | $3.1 \pm 2.5$ | $4.44 \pm 4.79$ | $26.97 \pm 6.11$ | $97.65 \pm 4.7$ | $1.61 \pm 0.29$ | $86.47 \pm 21.17$ | $100 \pm 0.0$ |
| Copy OOD (100-200) | $0.0 \pm 0.0$ | $0.0 \pm 0.0$ | $0.0 \pm 0.0$ | $0.0 \pm 0.0$ | $0.0 \pm 0.0$ | $66.04 \pm 40.49$ | $0.0 \pm 0.0$ | $40.89 \pm 31.27$ | $99.87 \pm 0.14$ |
| Copy OOD (200-300) | $0.0 \pm 0.0$ | $0.0 \pm 0.0$ | $0.0 \pm 0.0$ | $0.0 \pm 0.0$ | $0.0 \pm 0.0$ | $3.54 \pm 4.35$ | $0.0 \pm 0.0$ | $2.08 \pm 2.14$ | $98.16 \pm 1.82$ |

Results are in Table 1. We draw the following conclusions:

**Most PE methods struggle OOD:** Absolute and relative positional embeddings consistently struggle in out of distribution settings. Positional information is attached to specific indices. Best case they learn a bias in a particular direction (recency or the opposite), but even then this bias is input solely index dependent and only applicable for IID indices. Consequently, they struggle with OOD settings. RoPE also appears to learn a strategy which is fit to the training distribution and therefore struggles with out of distribution settings. This issue extends even to the differential transformer as it is based on RoPE. Consequently, despite having a more flexible selection mechanism it is still limited by its underlying positional encoding.

**NoPE Generalises Weakly:** The NoPE hypothesis (Kazemnejad et al., 2023) states that decoder only transformers do not need positional embeddings because they are able to approximate a counter. The theory is that they do so by selecting a token which serves as an anchor and sends some signal via the value message.

Intervening tokens send a null signal. The position of a given token is then determined by the extent to which the anchor token's signal is diminished by the softmax. From Table 1 we can see that this strategy generalises better than absolute, relative and rotary PE, consistent with the author's findings. However, it still does very poorly overall. NoPE fails at copy and induct indicating that when the anchor signal becomes too diluted due to increased length it fails to convey position accurately. The total failure of NoPE on the flip flop dense OOD set indicates issues when there are a number of possible anchor tokens in close proximity to one another. Too many anchor signals lead to errors in the count. Results show that NoPE is insufficient for true generalisation.

**Flexible PE Yields Improvements but is Insufficient:** The strongest contenders of the baselines utilise more flexible positonal encodings. Label, which randomly selects from a larger set of positions, is the weakest among them, but performs better than simply using absolute positions. However, alone the approach is insufficient for strong generalisation. FoT performs well on the copy and flip flop tasks, but is not able to fully generalise to either. Similar performance patterns are observed with CoPE. However, both methods struggle to generalise to the induct task which requires as its first step sharp attention to the predecessor token. This operation, also known as token shift (Peng et al., 2023), is purely positional and requires the a token to completely remove attention from itself and then apply a sharp recency bias to capture the predecessor. TRA can use its threshold mechanism to perform this operation while CoPE and FoT lack this capability and consequently suffer.

**TRA succeeds:** it is able to robustly generalise to all tasks. Its performance on copy and induct demonstrate strong capacity for memory and associative recall. Its ability to solve flip-flops shows that it can combine positional and semantic information in a complementary fashion. To the best of our knowledge it is the first transformer network to fully solve the flip-flop benchmark of Liu et al. (2023a). It achieves 100% accuracy across seeds, which till this point has been only been achievable by LSTM networks. However, both FoT and CoPE come very close to full generalisation. To further tease apart performance differences on this core capability we generate an additional synthetic test in the same spirit as the original benchmark.

### 3.4 Flip-Flops++

**Task:** To further test the differences between contextualised positions and the baseline approaches we create a further synthetic test which we refer to as Flip-Flops++. The task is a follows: the input consists of instruction followed by a sequence of random symbols combined with a trigger symbol. The set of instructions are as follows: return the symbol that occurs immediately after the *first occurrence* of the trigger, immediately after it, and immediately before or after the last occurence. For example, if we use a as the trigger symbol an example could consist of: (instruction: before first, sequence: bcxaklcaztyab, out: x). The training set contains examples from each of the instructions and sequence lengths of 0-50. The test set consists of of sequences of lengths 51-500. This is a more challenging scenario than the original benchmark as dependency differences can be starker than in the original sparse set, and the model is required to behave differently depending on the instruction. Moreover, because we include having to retrieve from the first occurrence among the instructions it lets us investigate whether techniques which employ a recency bias are limited to only being able to attend to the most recent relevant item. For training we employ the same hyperparameters as before.

Table 2: Length generalisation results by instruction on Flip-Flop++, taken over four random initialisations. Metric is full sequence accuracy (exact match).

| Model | NoPE | APE | REL | RoPE | Label | FoT | Diff | CoPE | TRA |
|---|---|---|---|---|---|---|---|---|---|
| After First Match | $64.53 \pm 19.36$ | $70.78 \pm 18.37$ | $95.78 \pm 5.1$ | $99.27 \pm 0.96$ | $96.80 \pm 3.91$ | $51.74 \pm 18.08$ | $92.01 \pm 2.07$ | $\mathbf{100 \pm 0.0}$ | $95.64 \pm 4.87$ |
| After Last Match | $20 \pm 4.0$ | $11.77 \pm 1.84$ | $25.16 \pm 6.09$ | $38.06 \pm 7.09$ | $37.58 \pm 10.98$ | $65.0 \pm 22.86$ | $27.74 \pm 1.82$ | $89.52 \pm 10.21$ | $\mathbf{99.84 \pm 0.28}$ |
| Before First Match | $58.23 \pm 18.15$ | $71.17 \pm 15.67$ | $86.32 \pm 21.67$ | $98.68 \pm 0.87$ | $87.94 \pm 12.47$ | $57.8 \pm 15.07$ | $93.53 \pm 2.39$ | $\mathbf{100 \pm 0.0}$ | $98.97 \pm 1.78$ |
| Before Last Match | $13.45 \pm 0.69$ | $10.12 \pm 2.24$ | $27.06 \pm 9.02$ | $33.86 \pm 15.61$ | $34.34 \pm 12.97$ | $76.42 \pm 13.72$ | $23.25 \pm 4.28$ | $91.3 \pm 6.32$ | $\mathbf{100 \pm 0.0}$ |

Results are shown in Table 2. As before most PE methods struggle to generalise, though RoPE performs quite strongly with first match, echoing the findings of Barbero et al. (2025). However, in this more challenging setting, only the contextualised schemes, CoPE and TRA, are able to robustly generalise. Though TRA performs better at last match while CoPE holds a slight edge at first match, due to their use of different

positional schemes. For natural language it is likely that the former is preferable as it better reflects causal structure (i.e. it allows understanding of an entity's *current* state). Notably FoT, which in the original Flip-Flop benchmark of Liu et al. (2023a) appeared to be a strong contender, suffers substantial performance degradation in this more challenging setting. This is because, as we hypothesised in the introduction, it is unable to synergise positional and semantic information and consequently fails to generalise to capability (3) from earlier.

### 3.5 Why does contextualised relative distance matter?

In order to analyse the impact of contextualised relative distance we train two layer one head TRA and FoT models on the flip-flop language modeling task and compare the attention heatmaps for the final layer in Figure 1. TRA fully generalises in this limited setting, the only model to be able to do so out of all the baselines, and consistent with the optimal solution demonstrated in theory by Liu et al. (2023a). On the other hand FoT fails, and the reason why is clearly visible by contrasting the attention weights of the two.

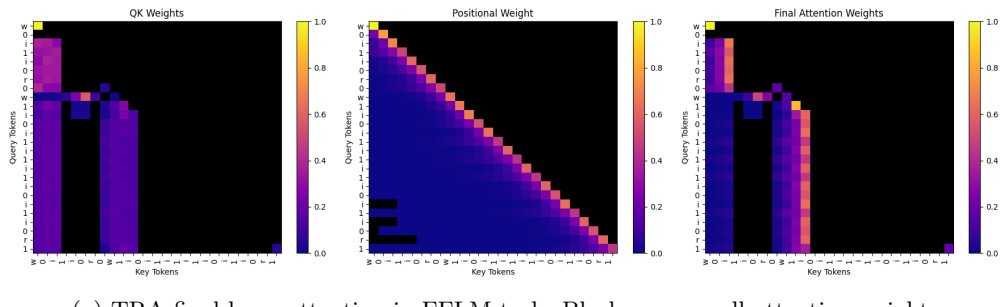

(a) TRA final layer attention in FFLM task. Black means null attention weight.

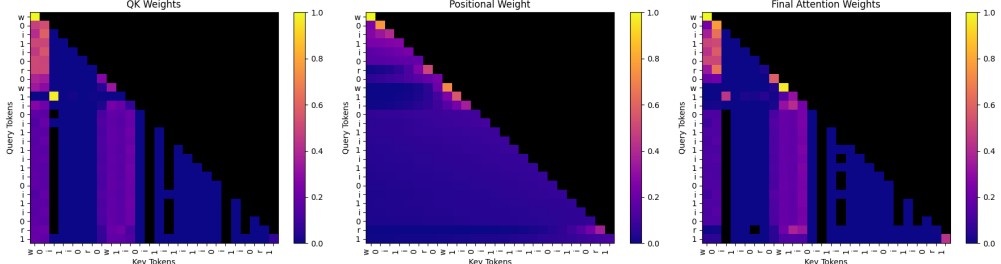

(b) FoT final layer attention in FFLM task. Black means null attention weight.

Figure 1: Contrasting Attention Heatmaps between TRA and FoT. TRA synergises semantic and positional information while FoT must trade one for the other.

Figure 1a shows how TRA is able to combine both semantic content with relative positions. It first filters out irrelevant keys fully and is then able to apply a strong positional bias to the remainder that truly matter, allowing it to cleanly select the most recent write instruction. Its recency bias (1a centre) is applied in a shifted manner only to what is relevant, which allows it express a strong positional preference. FoT on the other hand (1b), has no such mechanism and as a result must diminish its recency bias to be able to focus on semantic content (contrast the strength of 1b centre with 1a centre). This fact, coupled with its inability to completely remove irrelevant tokens, means that it is unable to handle long distance dependencies and consequently cannot learn semantic and positional preference such that they complement each other. We use FoT here as a clean illustrative example, but note that the same failure mode *must* be true of all schemes that do not employ contextualised distance.

### 3.6 Language Modeling:

TRA enables improved generalisation in controlled synthetic settings, which we attribute to its capacity for contextualised relative distance. What happens when we expose it to more complex data like natural

language? Investigating this question has two core motivations. Firstly, contextualised relative distance requires selective sparsity to operate. This operation cuts the connection between columns and we must make sure that this does not harm learning in more complex domains by potentially limiting exploration. Secondly, it allows us to investigate whether the generalisation patterns observed in synthesis also occur in natural language. For our experiments we turn to the WikiText-103 benchmark (Merity et al., 2016), which consists of full Wikipedia articles comprising circa 100 million tokens, probing both scale and ability to handle long distance dependencies.

**Hyperparameters:** For language modelling we increase model size to the medium configuration of Turc et al. (2019); Opper et al. (2023a). This is eight layers, eight heads and a 512 dimensional embedding size. The MLP is set 2x embedding size for both the linear and gate hidden units as before. We use the GPT-2 tokenizer (Radford et al., 2019) which leads to a vocabulary size of roughly 52k. Totaling circa 80 million parameters for both TRA and the baselines. We train using: window size 128, batch size 64, 100k steps. Our scheduling regime remains the same as before. Our evaluation consist of two parts. In distribution: we measure perplexity on the test set using the training window of size of 128. Out of distribution: we increase the window size to 4096 and observe the extent to which perplexity changes with increased length.

Table 3: Test Perplexity on WikiText-103 ($\downarrow$). Mean and standard deviation taken over four random initalisations. Models are trained for 100k steps with a context window of 128 tokens.

| Model | NoPE | APE | Rel | RoPE | Label | FoT | Diff | CoPE | TRA |
|---|---|---|---|---|---|---|---|---|---|
| Perplexity | $31.74 \pm 0.11$ | $31.61 \pm 0.16$ | $31.27 \pm 0.11$ | $30.11 \pm 0.04$ | $32.09 \pm 0.07$ | $30.30 \pm 0.04$ | $29.46 \pm 0.12$ | $30.20 \pm 0.09$ | $30.04 \pm 0.32$ |

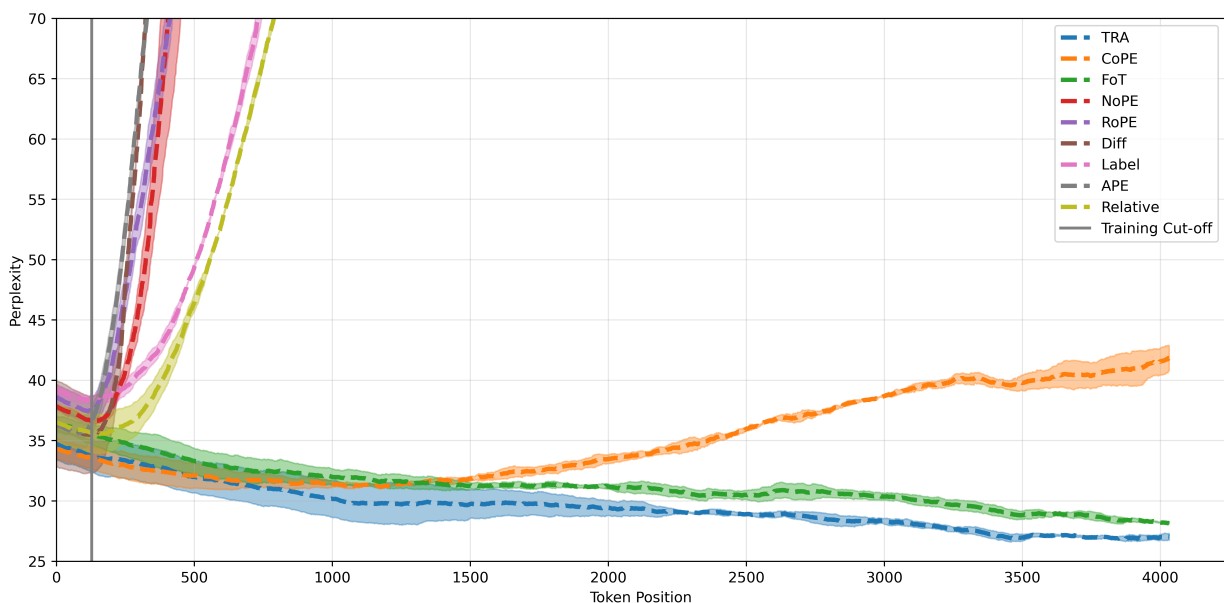

Figure 2: WikiText-103 Test Perplexity ($\downarrow$) on OOD sequences lengths. Models trained for 100k steps with a window size of 128, and evaluated with window size up to 4098. Results taken over four random seeds.

**Results:** In-distribution results are shown in table 3. TRA remains competitive with the strongest baselines. This means that we can introduce selective sparsity without hurting learning. Notably the strongest performing model *in-distribution* is the differential transformer (Ye et al., 2024), which contains a more sophisticated mechanism for relevancy determination. Replacing the simple ReLU we use here with such a more sophisticated approach could prove a fruitful avenue for future work. The out of distribution case is shown in figure 2. Most PE methods very rapidly begin to decline in quality, with similar degradation patterns emerging compared to the synthetic settings. The three chief exceptions to this rule are TRA, CoPE and FoT. CoPE is able to maintain/slightly improve perplexity for up to roughly a 10x increase in

context length, before performance begins to deteriorate. We believe this is because its use of fractional positions only temporarily alleviates generalisation difficulties, but ultimately leaves the core issue unaddressed. Furthermore, increasing the number of positions in CoPE may delay degradation further, but any increase comes with a heavy computational burden and consequently makes CoPE difficult to scale as the sole mechanism for representing position. On the other hand, both FoT and TRA are capable of strong length generalisation (up to 32x greater than training length). Moreover, they display a promising trend of improved perplexity with increased length which indicates that they are able to *make use of* increased context rather than simply being robust to it. Between the two, TRA demonstrates a consistent perplexity edge compared with FoT. We believe this is due to its improved capacity for handling out of distribution dependency lengths as demonstrated by the flip-flop language modeling results. In sum, we show that TRA can introduce selective sparsity to the attention weights without compromising performance, and that its improved generalisation capabilities in synthetic settings also appear to be mirrored in the more complex setting of natural language. Finally, to provide an intuition for how TRA prefers to use its attention and forget gate we provide an analysis the attention heatmaps learned by our language models in Appendix C.

## 4    Related Work:

**Generalisation in Neural Networks:** Neural networks, and particularly transformers, struggle with out of distribution generalisation (Hupkes et al., 2019; 2023). Partially this is architecture dependent. Significant efforts have been made towards creating architectures that enable a more flexible information flow and consequently have access to greater space of possible solutions (Fan et al., 2020; Chowdhury & Caragea, 2021; Opper et al., 2023b; Opper & Siddharth, 2024b; Soulos et al., 2024; DuSell & Chiang, 2024). However, these approaches tend to come at the cost of either flexibility or parallelisability - the key properties which made transformers, and particularly their decoder only variants, take over the world.

**Transformer Programs:** A parallel effort has been made to understand what kind of finite depth solutions transformers *should* be able to represent (Weiss et al., 2021; Friedman et al., 2023; Smolensky et al., 2024). One motivation being the conjecture that transformers will learn to generalise iff they are able to represent a program which does so (Zhou et al., 2023). However, empirical validation for this conjecture has thus far remained limited. Even when we expect the models to generalise they often only do so weakly, and rigorously bridging the gap between theory and practice remains a long standing challenge. A key motivation for TRA came from the theoretical model presented by Smolensky et al. (2024) which draws analogy between the transformer and the classical production system architecture from cognitive science. Their theoretical model has the attention mechanism first exactly match on specified conditions before applying a hard positional tie-breaker to perform one-hot selection. TRA arose out of an ongoing effort to create a learnable equivalent to their theoretical model.

**Length Generalisation in Transformers:** Several mechanisms have been posisted to improve transformer length generalisation. We partition this work into two categories. The first deals with trying to learn positional weights that are flexible w.r.t. the length distribution of the training data (Dubois et al., 2019; Ruoss et al., 2023; Li & McClelland, 2023; Ray Chowdhury & Caragea, 2023; Golovneva et al., 2024). Orthogonally, recent work has demonstrated that when extending to longer sequences the softmax tends towards uniformity and therefore introduces errors the attention signal (Ye et al., 2024; Veličković et al., 2024). TRA investigates whether these two streams of research should combine. It attempts to reduce noise in the attention through selectively sparsity and learns general positional encodings with its forget gate.

**Sparsifing Attention:** Finally, a substantial amount of effort has been made towards introducing sparsity into attention. One vein attempts to reduce the quadratic cost of the attention operation by applying fixed patterns (Child et al., 2019; Zaheer et al., 2021; Nawrot et al., 2025), though doing so either requires restricting flexibility or is only applicable during inference time. Another has sought to allow sparsity within the softmax by allowing for zero probabilities (Peters et al., 2019), however, such algorithms often depend on computational complex operations and have delivered mixed results. Finally, Zhang et al. (2021) replace the softmax with ReLU, thereby introducing selective sparsity and demonstrating promising results on NMT.

## 5   Conclusion

In this work we investigate whether fundamental issues with length generalisation occur due to a conflict between positional and semantic information. As a probe for this hypothesis we present TRA, a simple modification to the attention mechanism designed to allow both types of information to operate in tandem. We find that the introduction of such a change significantly improves out of distribution generalisation. Moreover, the core innovation of TRA comes from marrying selectively sparsity with contextualised relative distance. We see substantial opportunity for development of future architectures along this axis. Selective sparsity allows columns to focus on what's important, while the forget gate enables a memory over past time steps. While this latter mechanism does not enable full state tracking behaviour, temporal memory mechanisms have shown enhanced ability to approximate $NC^1$ solutions compared with standard transformers (Merrill et al., 2024). This means that combining a more sophisticated memory with a more flexible sparsification threshold should yield models which are able to approximate complex algorithmic reasoning more cleanly and with fewer layers. Pursuing research in this direction has the promise of bridging the gap between our theoretical notions of what transformers should be able to do and observed empirical realities, and in so doing yield models which are capable of learning more robust, generalising attentive circuits.

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

## A  Implementation:

```python
class TRA(nn.Module):
    def __init__(self, e_dim, n_heads, dropout=0.01):
        super(TRA, self).__init__()
        self.E = e_dim
        self.e = e_dim // n_heads
        self.h = n_heads
        self.tokeys =  nn.Linear(e_dim, e_dim, bias=False)
        self.toqueries = nn.Linear(e_dim, e_dim, bias=False)
        self.tovalues = nn.Linear(e_dim, e_dim, bias=False)
        self.delta_proj = nn.Linear(e_dim, self.h)
        self.c_out = nn.Linear(e_dim, e_dim)
        # use qk norm but no scale to save params
        self.qk_norm = RMSNorm(self.e, elementwise_affine=False)
        self.register_buffer("bias", torch.triu(torch.ones(max_len, max_len),
    diagonal=1).view(1, 1,max_len, max_len).to(torch.bool))
        self.dropout = nn.Dropout(dropout)

    def forward(self, x):
        b, s, _ = x.shape # batch_size, seq_len, e_dim
        # transform q, k, v
        q = self.toqueries(x).view(b, s, self.h, self.e).transpose(1, 2)
        k = self.tokeys(x).view(b, s, self.h, self.e).transpose(1, 2)
        v = self.tovalues(x).view(b, s, self.h, self.e).transpose(1, 2)
        # qk norm (no scale -> can share)
        q = self.qk_norm(q)
        k = self.qk_norm(k)
        # compute attn dot
        S = (q @ k.transpose(-2,-1)) / torch.sqrt(torch.tensor(self.e, dtype=torch
    .float, device=x.device))
        S = S.masked_fill(self.bias[:,:,:s,:s], float('-1e11'))
        S = F.relu(S)
        mask = (S == 0)
        # compute positional weights D
        delta = F.logsigmoid(self.delta_proj(x)).transpose(1,2).unsqueeze(-1).
    contiguous().view(b, self.h, s, 1)
        D = (~mask).float() * delta
        D = D.flip(-1).cumsum(-1).flip(-1)
        # dropout crucial to avoid loss spikes in language modelling
        A = self.dropout(S + D)
        A[mask] = -1e11 # mask: casual + threshold
        A = A.softmax(dim=-1)
        # protect against leaks where mask all = 0 from no-op attention
        A = A.masked_fill(mask.all(-1).unsqueeze(-1), 0)
        # compute output
        out = A @ v
        out = out.transpose(1, 2).contiguous().view(b, s, self.E)
        return self.c_out(out)
```

Listing 1: TRA Implementation

## B  Hyperparameters:

For synthetic tasks we train for one full pass through the data, for copy and induct this required circa 100k steps as our training set consisted of 4 million examples, while for flip-flop language modeling this corresponded to 20k. We used batch size 128 for the former tasks and batch size 64 for FFL to accomodate its larger sequence length (512).

For RoPE we set the theta to 500k following Dubey et al. (2024). For CoPE we set npos max to 64 following the original authors (Golovneva et al., 2024). All models are trained using the AdamW optimiser with a low dropout of 0.01 applied to the attention weights and feedforward hidden representation.

## C  Attention Analysis: Language Modeling

Trained TRA models exhibit both completely content focused and completely positional attention heads, as well as ones where the trade-off is somewhere in between. As a general rule of thumb the more sparsely activated the QK content weights, the weaker the degree of time decay enforced by the forget gate. Conversely, dense QK activations lead the model to focus on a local window (see Figure 3 below). Attention patterns can be highly sparse indicating significant threshold pruning. Secondly, tokens can exhibit considerable variance in time decay, impossible with fixed schemes such as ALiBi Press et al. (2021).

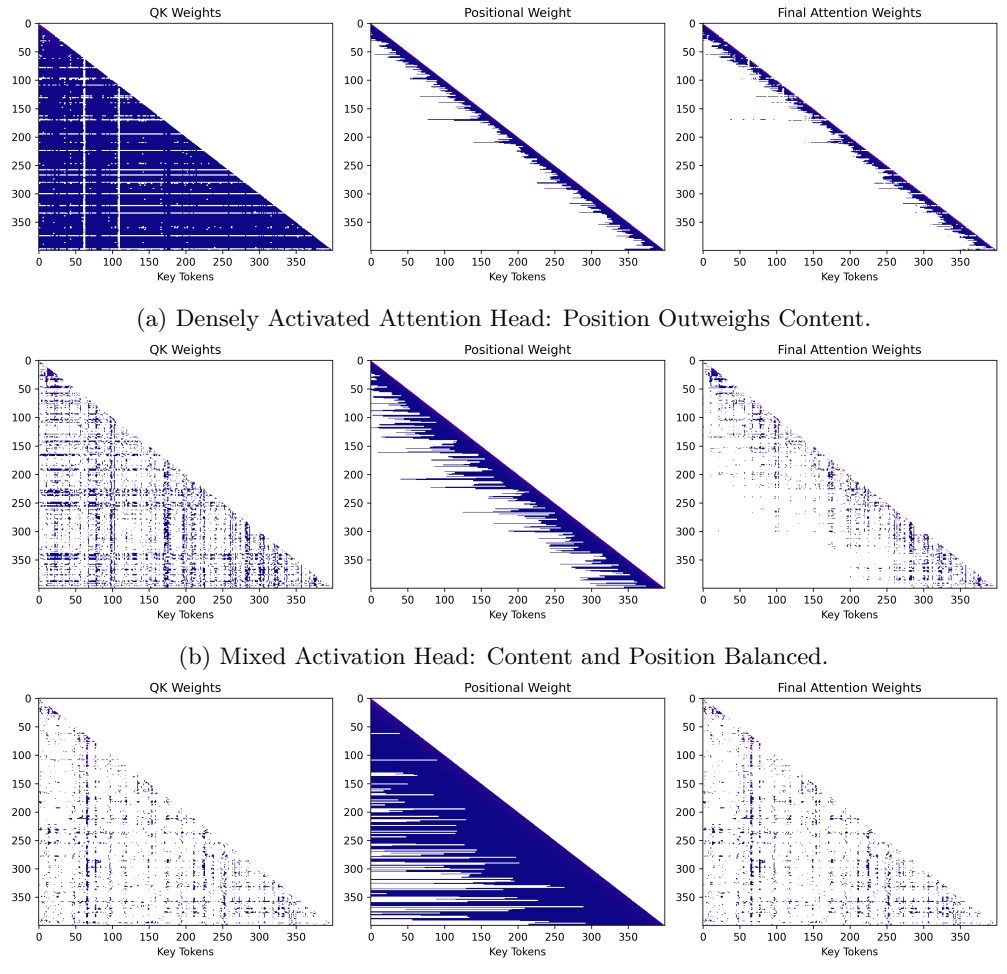

Figure 3: Attention Heatmaps from trained TRA language model. White signifies null attention.

## D Computational Efficiency and Potential Optimisations:

One potential disadvantage of TRA compared with schemes such as RoPE, which remain the current defacto choice for positional embeddings, is that it introduces additional operations which can diminish runtime efficiency. To give an indication across two configurations:

Tiny: 4 layers, 4 heads, residual stream size 256, ffn 1024
Medium: 8 layers, 8 heads, residual stream size 512, ffn 1024

On a single Nvidia A40 card with window size 256 and batch size 64:

Tiny RoPE: 20k steps 58.73 minutes
Tiny TRA: 20k steps 65.88 minutes (circa seven minute increase)

Medium RoPE: 20k steps 156.34 minutes
Medium TRA: 20k steps 178.19 minutes (circa 22 minute increase)

However, these differences are without any effort towards runtime optimisation on our end. We believe there is significant scope for even greater efficiency. Though this is currently highly speculative, and hence left out of the main paper, there is no reason why the TRA algorithm could not be computed in a tiled manner and take advantage of the optimisations introduced by flash attention Dao et al. (2022); Dao (2023). While not trivial because the blocks we need to be processed in a causal manner to allow for online computation of the cumsum, the fact that attention patterns can be highly sparse and the forget gate can enable early stopping of row processing means that there should be considerable scope for greatly improving the efficiency of TRA.

