# OpenReview forum: "TRA: Better Length Generalisation with Threshold Relative Attention"
_TMLR — Accepted by TMLR_

### Review · Reviewer_Zo6s · 2025-04-15

**Summary Of Contributions:**

The paper tackles the problem of length generalisation in attention mechanisms. In particular, they aim to address two known issues: 1) attention weights can never be 0 and noise therefore accumulates and 2) that positional encodings can "fall out of distribution" if they are calibrated on shorter context, thus harming length generalisation.

To tackle this problem, they propose a new attention mechanisms they call Threshold Relative Attention (TRA). TRA introduces a form of sparsity through the application of a ReLU function and a subsequent masking mechanism. It introduces a positional bias then by performing an interesting cumulative-sum type of operation which aims to behave like a learnable bias.

They evaluate this method on a number of synthetic benchmarks and on language-modeling.

**Audience:**

Yes

**Broader Impact Concerns:**

None that I am aware of.

**Claims And Evidence:**

Yes

**Requested Changes:**

- Formatting issues with Eq.5 and Eq. 6. It seems like there are missing brackets over the matrices.

- Add algorithm/pseudo-code of TRA (see weaknesses) in the Appendix.

- Add runtimes of TRA, comparing to RoPE (see weaknesses). I am expecting TRA to run slower, but it might be worth bringing this up as a limitation if indeed the runtime is (significantly) slower.

- As the work of Barbero et al. (2025) is repeatedly mentioned, it might be useful for completeness to add their method of p-RoPE to Table 1 as it aims to address length generalisation issues of RoPE to some extent. I see this as an optional change as there are already a number of comparisons to different PEs.

- Could the authors report typical values of $\delta_i$ in Eq. 8 in a trained model. As the function decays exponentially fast, this will quickly go to 0 as D grows. It would be interesting to see how the model prefers to use this mechanism.

**Strengths And Weaknesses:**

Strengths

- The work is well positioned and highly relevant to recent literature
-  The proposed method is unique and its strengths are clear.
- There are a number of interesting synthetic experiments that showcase the strength of the method.
- The attention maps in Figure 1 are an interesting addition and show the interpretability of the technique.


Weaknesses

- There is no discussion of the run-time of the method compared to RoPE. RoPE is particularly useful because it can be very easily optimised due to its 2x2 block-diagonal nature. It seems to me that these operations might slow down the forward and backward passes as they cannot be perhaps fused as easily. I believe it would be worth dedicating some space in the appendix to discuss this as a potential limitation (if it is indeed slower).
- The way TRA is presented in Section 2 is slightly confusing as there are a lot of different symbols. It took me a while to parse it. For instance I don't believe there is a need to introduce the intermediate $\hat{D}$ as one could simply write M directly.
- For additional clarity, it would be useful to place in the appendix the pytorch/Jax code or pseudo-code for the implementation of TRA for completeness.
- It would be interesting to see some results showing how for example there are configurations of TRA that can provably generalise over certain (synthetic) tasks while for instance RoPE cannot. At the moment the paper provides no theoretical insights which is fine, but this would help strengthen the work.

---

> ### Author Response · Authors · 2025-05-02
>
> Thank you for your positive appraisal of our work, and insightful follow up questions!
>
> **Runtime Differences**
>
> TRA is slower than RoPE though we do not think the increase is intractable. To give an indication across two configurations:
>
> Tiny: 4 layers, 4 heads, residual stream size 256, ffn 1024
> Medium: 8 layers, 8 heads, residual stream size 512, ffn 1024
>
> On a single Nvidia A40 card with window size 256 and batch size 64
> Tiny RoPE: 20k steps 58.73 minutes
> Tiny TRA: 20k steps 65.88 minutes (circa seven minute increase)
>
> Medium RoPE: 20k steps 156.34 minutes
> Medium TRA: 20k steps 178.19 minutes (circa 22 minute increase)
>
> However, these differences are without any effort towards runtime optimisation on our end. We believe there is significant scope for even greater efficiency. Firstly, we know that a cuda optimised version of FoT has been released, compatible with flash attention, which allows for considerable training speed. This should be easily adaptable to TRA as we would only need to add a ReLU (very inexpensive elementwise operation) a mask calculation (again inexpensive) and use that to simplify the cumsum. As a further point we have discovered an extremely recent paper from the FoT authors which is able to consistently reduce the number of FLOPs required for the softmax by 70% and increase training throughput by 10-35%. This is done by disregarding keys which have been decayed to the point of irrelevance by the forget gate [1]. TRA should be able to further improve upon this already impressive reduction because it discounts even more keys through its thresholding mechanism.
>
> **Notation**
>
> Thank you for pointing this out, we have amended section 2 following your feedback.
>
> **Reference Implementation**
>
> Good point - we have added a PyTorch implementation as an Appendix and will of course accompany this with a full code release should the paper be accepted.
>
> **Theoretical Insights**
>
> We can attempt a formal proof but would like to take some time to carefully consider this to make sure it is correct.
>
> However, the basic intuition as to why RoPE can’t perform contextualised distance style operations should hopefully be clear enough.
>
> The RoPE transformation is applied before the QK dot product. This means that it cannot be contingent on the results of said dot product. If we have understood Barbero et al correctly what it can do is:
>
> Purely positional operations by setting the keys to be identical and therefore just focussing on the rotation.
> Purely content based by focussing on the low frequencies, which are more invariant to relative distance.
>
> However, content based selection which then determines positional operations is firmly out of reach.
>
> Do you agree with this intuition? We would be interested to hear your thoughts!
>
> **P-RoPE**
>
> We will try to add this if we have time during the discussion period, however, we would like to prioritise some other reviewer requests as you have flagged it as optional. We found a jax implementation which we will have to translate to Pytorch and then run all our experiments so this may not be possible, but will do our best to accommodate it if we can!
>
> **Typical Delta Values**
>
> This is a good request and we think it should be included, unfortunately due to losing access to the original cluster since the time of the submission we don’t have saved checkpoints available. We are currently rerunning and will analyse results over the weekend to update our response by Monday.
>
> Thank you again for your feedback and time taken with the paper, we look forward to engaging with you during the discussion period!
>
> [1] https://arxiv.org/abs/2504.06949

---

> ### Author Response · Authors · 2025-05-06
>
> **Typical Delta Values**
>
> We have added an Appendix containing some analysis of the interplay between content and position in our trained language models. In densely activated heads, where attention is more uniform and few keys are excluded via the threshold, the forget gate plays a strong role, focussing attention on a local window. In sparsely activated heads, where most keys are completely removed via the threshold, positional bias is diminished and the attention provides a more global view. Then there are of course heads which occupy an intermediary role, focussing on longer horizons but still exhibiting some degree of sparsity, here the two mechanisms act together. Finally, while particular heads are generally focussed on either local or global information, the particular degree of decay introduced is dependent on per token delta values, and these can show some variance, indicating a local global trade-off on both the token and head levels.
>
> If there is any further analysis you would like please let us know and we will happily accomodate your request!

---

> > ### Comment · Reviewer_Zo6s · 2025-05-07
> >
> > Thank you for your response.
> >
> > *Runtimes*
> >
> > Thank you for providing the run-times. Are you planning on including them in the paper? For transparency, I believe it would be useful to add this discussion in the appendix. Hinting to the fact that TRA could further be optimised would also be good as well.
> >
> > *Theory*
> > After further thought, I think it is fine to avoid theoretical treatment in this work, as it should be quite clear what the advantages are of the technique due to the special type of masking. Regarding RoPE, yes I believe the main critique is that the slow rotations still accumulate in long context settings and this has the potential to break certain mechanisms.
> >
> > *Delta values*
> > Thanks for providing additional information. The plots are very interesting and I think provide nice evidence that the model is able to construct heads that trade-off in different ways position and content.
> >
> > Thanks for your efforts in the rebuttal. I believe the new results help to strengthen the paper.

---

> > > ### Author Response · Authors · 2025-05-07
> > >
> > > We have just added an amended version of the manuscript with a new appendix (D) discussing runtimes along the same lines as our response.
> > >
> > > Thank you again for your time and helping us to improve our paper!

---

### Review · Reviewer_DfmF · 2025-04-20

**Summary Of Contributions:**

This paper studies one of the main limitations of transformer models: their poor generalization to sequence lengths not seen during training. The authors identify two key failures in the standard self-attention mechanism that contribute to this problem, the inability to properly remove irrelevant information from the attention distribution and the issue that positional biases can unintentionally up-weight irrelevant tokens when dealing with out-of-distribution sequence lengths
To solve these issues, the authors propose Threshold Relative Attention (TRA), which introduces two main modifications to the standard attention mechanism:
- selective sparsity, to completely removing irrelevant keys from the attention softmax by applying a threshold to raw attention weights;
- sontextualised relative distance, that means only calculating relative positional information between the query and the keys that pass the relevance threshold.

The authors implement this by first applying a ReLU to the QK^T dot product to eliminate irrelevant keys, then computing a contextual relative distance only among the remaining relevant tokens, and finally using a parameterized forget gate to create a temporal decay effect.

The approach is evaluated on synthetic tasks designed to test length generalization as well as on the WikiText-103 language modeling benchmark. Results show that TRA outperforms various positional encoding methods  in out-of-distribution settings, achieving almost perfect scores on synthetic tasks even when sequence lengths are up to 6x longer than those seen during training.

**Audience:**

Yes

**Claims And Evidence:**

Yes

**Requested Changes:**

- Please work on the statistcal analysis of the experiments (as reported in weaknesses).
- Plase also add an analysis on the threshold value sensitivity and if that introduces new hyperparameters.
- I'd add connections to other sparse attention approaches and how TRA differs from those, and especially i'd underline how TRA is different from the paper "Lin, Zhixuan, et al. "Forgetting Transformer: Softmax Attention with a Forget Gate." arXiv preprint arXiv:2503.02130 (2025)" that is very recent, but i can see a possible overlap, so i'd clarify properly on that to underline the value of TRA.
- Finally, I'd add an analysis of training curves and convergence speed compared to baselines.

**Strengths And Weaknesses:**

Weaknesses:
- The paper doesn't analyze how sensitive the approach is to the threshold value or whether this introduces new hyperparameters that need tuning.
- Completely masking out tokens based on a threshold could potentially lose valuable information in more complex settings.
- The paper presents the results without a proper statistical analysis and some results appear to be from just 4 random initializations, which may not be enough for proper conclusions (for example: in table 1 they shows results on the Flip-Flop tasks and TRA achieves 100% accuracy while CoPE achieves 95.1% $\pm$ 4.4%. The standard deviation suggests that in some runs, CoPE might have achieved close to 99.5%. Without formal statistical testing, it's harder to be confident about how reliable this difference is.
And also, in the language modeling results, some perplexity differences between methods are quite small and without statistical significance testing, it's difficult to know if these small differences represent genuine advantages or just random variation)


Strenghts:

- I think the paper focuses on a core problem for the field of positional encodings and long length generalization and i really appreciate the approach the authors propose.

PS. I pointed out more weaknesses then strenghts because i think you can work on those few points to make your paper even better, i do like the core approach and i think it has a value.

---

> ### Author Response · Authors · 2025-05-02
>
> Thank you for your feedback, time taken reading the paper and helpful review!
>
> **Threshold Sensitivity and Hyperparemters:**
>
> We tried a few alternatives for computing the threshold, including GeLU and some more complicated approaches, however, it turned out that simply using ReLU was the most effective for now. Fortuitously, because ReLU is a non-linearity this doesn’t introduce any hyperparameters. We do think there is considerable potential in making the threshold value learnable, which would introduce potential complexities and maybe hyperparameters, but we leave that to future work.
>
> **Thresholding Impact on Complex Settings:**
>
> We agree that this is certainly a valid concern, which is why we introduce the language modelling evaluation alongside our synthetic experiments. Language modelling requires taking into account a lot of complex interactions and the LLM era demonstrates just how much complexity learning to predict the next word can entail. A further note is that prior work [1] replaced the softmax in transformers with ReLU followed by RMS norm for resource efficiency purposes and demonstrated strong performance in machine translation, another highly complex task. Which lends some external validation to our findings. Otherwise, if there is a particular task you have in mind let us know and we will do our best to incorporate it during the discussion period!
>
> **Statistical Testing**
>
> You are correct to point out that in some cases the difference is small, so we would like to clarify them.
>
> CoPE on the flip-flop language modelling benchmark:
>
> Firstly, the key common feature that CoPE and TRA share is that they both use contextualised distance schemes and as such we fully expect both models to do well on the flip flop task which explicitly requires this form of reasoning. The chief benefits of TRA as compared with CoPE is that it is significantly more computationally efficient and does not suffer the limitations of a fixed set of positional encodings. Where we do see a clear benefit is in the out of distribution language modelling case (Figure 2) which brings out the limitations of CoPE.
>
> Secondly, to pass the original flip-flop benchmark, models need to consistently generalise across seeds. The authors of the benchmark [1] show that while LSTM-based networks can easily do this, transformers consistently exhibit a long tail of generalisation failures, however slight, and no matter how much they increase scale or tweak the architecture,they are unable to bridge this gap. TRA is able to succeed and always does so over the hundreds of runs we tested during its development. Moreover, we can restrict the model to two layers/one head and still achieve perfect generalisation. We did some mechanistic analysis of the solution it learned and are able to show that it learns the provably optimal circuit shown in theory by Liu et al [1]. While we omitted this from the paper because we thought it would excessively belabour the point, we would be more than happy to add this analysis in an Appendix if you think it would be of interest!
>
> In-distribution language modelling:
>
> The key point of the in-distribution setting is to demonstrate that TRA is not significantly worse (i.e. that the changes we made don’t hurt). The difference between approaches is generally minor and we don’t claim that TRA is superior in this setting. Where it does shine is in the out-of-distribution setting where we can clearly see massive perplexity spikes from most baselines while TRA holds steady/improves.
>
> **Greater Emphasis on Differentiation to FoT**
>
> Good point - it can’t hurt to emphasise our contribution and the difference to this key baseline. We have highlighted this in the description of FoT at the start of sec 3, and have added a passage to the related work section outlining various prior sparse attention mechanisms.
>
> To recap, the crucial difference is that FoT lacks the capacity for contextualised distance. This means that if it needs to handle long-distance dependencies it must completely deactivate its positional weights, which results in the generalisation failures on synthetic tasks that we show in our first two experiments and in Figure 1.
>
> **Training Curves**
>
> Unfortunately, we have lost access to the original cluster so we do not have the curves to hand, but we did not observe anything particularly notable. On the synthetic tasks all models converged to their best values within the step count and were not helped by increase. For language modelling we kept the step count fixed but saw the same pace of convergence and IID performance, as you noted. We can rerun our experiments and if we find something of note we will include it in an appendix and update our response.
>
> Thank you again for your time taken to help us improve the paper. If you have any further questions please do let us know, and we look forward to engaging with you during the discussion period!
>
> [1] https://arxiv.org/pdf/2104.07012

---

### Review · Reviewer_YN4o · 2025-04-23

**Summary Of Contributions:**

This paper revisits positional encoding by incorporating richer contextual information derived from the query-key (QK) dot product. The approach begins by constructing a boolean mask that identifies relevant positions based on QK interactions. This mask is then used to compute a contextualized relative distance, which is parameterized and added to the attention weights prior to the softmax operation. The proposed method is evaluated against various positional encoding schemes—such as Absolute, Relative, and RoPE—across multiple benchmarks, consistently demonstrating improved performance. I particularly appreciate Figure 1, which effectively illustrates how the method integrates contextual position information into the final attention computation.

**Audience:**

Yes

**Broader Impact Concerns:**

I have no major concerns regarding the broader impact of the proposed method. It presents a clear technical contribution as an enhancement to positional encoding, focusing on methodological improvements.

**Claims And Evidence:**

Yes

**Requested Changes:**

I would appreciate it if the authors could address the concerns outlined in the Weaknesses section—particularly by providing a more thorough justification of the proposed scheme and including formulation-level comparisons with existing positional encoding methods.

I also have a few minor suggestions:
- The transition from Equation (5) to (6) is unclear. Including more detailed explanations and intermediate steps would help readers better understand the underlying mechanism.
- In Section 2.3, the variable $E$ is introduced without definition. Please clarify its meaning. Additionally, "ith" should be formatted as "$i$th" for consistency with mathematical notation.
- The usage of "1e11" is somewhat opaque. While I understand it is intended as a large negative value to suppress attention weights (by pushing them towards zero after the softmax), a more explicit explanation would be helpful, especially for readers less familiar with this technique.

On a positive note, the experimental sections are clearly presented and easy to follow—I have no issues with their clarity or structure.

**Strengths And Weaknesses:**

Strengths
- Introduces a novel positional encoding strategy that leverages contextually relevant positions informed by the QK dot product.
- Demonstrates strong empirical performance across a range of benchmarks, outperforming several established positional encoding methods.
- Provides clear and insightful analyses that help identify and interpret the relevant positional relationships captured by the model.

Weaknesses
- Section 2: The rationale behind the proposed framework is not sufficiently clear. Specifically, the design choices for computing contextual relative distances, their parameterization, and their integration into attention weights seem underexplained, given the multiple possible alternatives.
- The formulation of the proposed method lacks contrast with existing approaches. The paper would benefit from a side-by-side comparison: presenting the formulation of a standard positional encoding method and directly juxtaposing it with the proposed approach to clarify what is novel and different.

---

> ### Author Response · Authors · 2025-05-02
>
> Thank you for your review, helpful feedback and positive appraisal of our work!
>
> In order to address your concerns we have rewritten section 2 following your suggestions. We hope that you find this sufficiently clarifies things and incorporates your helpful suggested changes. In addition, we would also like to use our response to further highlight two central points, which we have also reinforced in our updated manuscript, regarding the rationale for TRA and how it is implemented:
>
> **Why should position be contextual?**
>
> Consider a case where you want to track the state of an entity over time, e.g. when you need to know the activity of a particular character in a book in order to predict what might happen next. A clean generalising circuit that solves this could select all keys that match that character and then pull information from the most recent one. This means that you first need to match relevant keys before applying your positional bias. Looking at the prevailing techniques, we see that they can’t perform this action cleanly:
>
> APE: adds positional information to the residual stream at layer zero entangling it with the other semantic content and making things difficult to separate; also limited due to fixed PE matrix size
>
> RPE: Positional weight is completely independent of the key it is applied to and added to the QK weights, so you cannot form the circuit; also suffers from a fixed size of weights
>
> RoPE: applies its transformation before the dot product is calculated, which again entangles the two pieces of information preventing clean formation of the circuit
>
> Our hypothesis in the paper (see penultimate paragraph of the introduction) is that this entanglement explains certain generalisation failures in transformers and that’s why TRA, which allows for contextualised positions, generalises better than the baselines.
>
> **Why parameterise with a forget gate?**
>
> The intuition is that the core purely positional operation that transformers make use of is contextualising within a local window, allowing for situating computation within the immediate context. Prior work [1] showed that simply hardcoding a recency bias already does really well. However, the downside is that it is hardcoded, which means that when you want something other than the local window you are unfortunately stuck. Parameterising the operation lets you learn an adjustable window size or to completely switch off the bias if necessary. Secondly, applying a constant decay removes the headache of a fixed set of weights that have to be capped when you exceed them. Thirdly, constant time decay is provably an extremely useful property. This is how state space models attempt to mimic true state tracking behaviour, which lets them approximate circuits of greater depth than baseline transformers and consequently generalise more easily [2]. Introducing the same mechanism to transformers, which are much more flexible because they don’t have to compress the whole sequence history into a single vector, seems like the natural next step if we want models that generalise better.
>
> Do let us know if the edits address your concerns or if you have any further follow up questions. Thanks again for your time with the review and we look forward to engaging with you during the discussion period!
>
> [1] https://arxiv.org/abs/2108.12409
>
> [2] https://arxiv.org/abs/2404.08819

---

### Decision · Action_Editor_DeQd · 2025-05-27

**Recommendation:** Accept as is

**Comment:**

This work presents a very insightful modification to prominent attentional mechanisms, with a special emphasis on an important downstream problem (improved length generalisation). All reviewers are in agreement that the work clearly meets (and exceeds!) the bar for publication in TMLR. Well done!

**Audience:**

Yes, it is evident that the topic at hand (length generalisation in Transformers) will be highly important to LLM practitioners and researchers, particularly taking into account the increasing emphasis on long-horizon modelling.

**Claims And Evidence:**

Yes, after the rebuttal all reviewers clearly agree that the work should be accepted.